# Malnutrition, Sarcopenia, and Malnutrition Sarcopenia Syndrome in Idiopathic Pulmonary Fibrosis

**DOI:** 10.3390/arm93030011

**Published:** 2025-05-29

**Authors:** Eva Cabrera-César, Rocío Fernández-Jiménez, Javier Lopez-Garcia, Alicia Sanmartín-Sánchez, Miguel Benítez Cano-Gamonoso, Isabel Asschert Agüero, Francisco Espíldora-Hernández, Luis Fernandez de Rota Garcia, Isabel Vega-Aguilar, Maria del Mar Amaya-Campos, Francisco J. Tinahones, Jose Manuel Garcia-Almeida, Jose Luis Velasco-Garrido

**Affiliations:** 1Department of Neumology, Virgen de la Victoria University Hospital, 29010 Málaga, Spain; javi.medicina.uma@gmail.com (J.L.-G.); michibcg@gmail.com (M.B.C.-G.); isabel.asschert88@gmail.com (I.A.A.); lfderota.neumologia@gmail.com (L.F.d.R.G.); jlvelascogarrido@hotmail.com (J.L.V.-G.); 2Department of Endocrinology and Nutrition, Virgen de la Victoria University Hospital, 29010 Malaga, Spain; isabel.mva13@gmail.com (I.V.-A.); mariadelmarac2@gmail.com (M.d.M.A.-C.); fjtinahones@hotmail.com (F.J.T.); jgarciaalmeida@gmail.com (J.M.G.-A.); 3Instituto de Investigación Biomédica de Málaga y Plataforma en Nanomedicina-IBIMA Plataforma BIO-NAND, 29010 Malaga, Spain; 4Department of Medicine and Dermatology, Málaga University, 29016 Malaga, Spain; 5Department of Endocrinology and Nutrition, Quironsalud Málaga Hospital, Av. Imperio Argentina, 29004 Malaga, Spain; 6Department of Endocrinology and Nutrition, Son Espases Universitary Hospital, Carretera de Valldemossa, 07120 Palma, Spain; alicia.sanmartin@outlook.es; 7Department of Neumology, Carlos de Haya Regional University Hospital, 29010 Malaga, Spain; fespildorahernandez@gmail.com; 8Department of Endocrinology and Nutrition, Hospital Universitario Virgen de la Victoria, CIBEROBN, Carlos III Health Institute (ISCIII), University of Málaga, 29016 Malaga, Spain; 9Centro de Investigación Biomédica en Red Fisiopatología de la Obesidad y Nutrición (CIBEROBN), Carlos III Health Institute (ISCIII), University of Málaga, 29010 Malaga, Spain

**Keywords:** idiopathic pulmonary fibrosis, malnutrition, sarcopenia, GLIM, EWGSOP2

## Abstract

**Highlights:**

**What are the main findings?**
The prevalence of malnutrition and sarcopenia in idiopathic pulmonary fibrosis (IPF) patients was assessed using GLIM and EWGSOP2 criteria.77.65% of patients were malnourished, 20% had sarcopenia, and 8.23% presented with the combined malnutrition-sarcopenia syndrome.Malnourished patients had significantly lower body weight, height, and muscle mass, as well as poorer health-related quality of life scores.Muscle mass was measured using bioelectrical impedance analysis (BIA), and sarcopenia was screened based on international guidelines.

**What is the implication of the main finding?**
The study highlights the need for systematic screening for malnutrition and sarcopenia in IPF to improve comprehensive patient management.

**Abstract:**

**Introduction:** Idiopathic pulmonaryy fibrosis (IPF) is a progressive interstitial lung disease with a poor prognosis. While comorbidities like pulmonary hypertension and lung cancer have been studied extensively, less attention has been paid to the implications of malnutrition and sarcopenia in patients with IPF. This study aimed to assess the prevalence of malnutrition, sarcopenia, and the combined malnutrition-sarcopenia syndrome in patients with IPF using the latest diagnostic criteria from the Global Leadership Initiative on Malnutrition (GLIM) and the European Working Group on Sarcopenia in Older People 2 (EWGSOP2). **Methods:** A prospective, observational, multicenter study was conducted, focusing on patients with idiopathic pulmonary fibrosis (IPF). All participants provided informed consent, and the study followed ethical guidelines. Malnutrition was diagnosed based on the GLIM criteria, requiring one phenotypic and one etiological criterion, with muscle mass assessed via bioelectrical impedance analysis (BIA). Sarcopenia was screened following the EWGSOP2 recommendations. The statistical analysis was performed using JAMOVI version 2.3.22, with significance set at *p* < 0.05. **Results:** The findings revealed that 77.65% of the participants were malnourished, and 20% had sarcopenia. The malnourished patients had significantly lower body weight, height, and muscle mass compared to the non-malnourished patients. Furthermore, the patients with malnutrition exhibited poorer health-related quality of life scores. This study also identified the malnutrition-sarcopenia syndrome in 8.23% of the participants. **Conclusions:** Malnutrition, based on the GLIM criteria was identified in three out of four patients with IPF, while sarcopenia according to the EWGSOP2 was present in one out of five. This study underscores the necessity for routine screening for malnutrition and sarcopenia in patients with IPF.

## 1. Introduction

Idiopathic pulmonary fibrosis is an interstitial fibrosing disease with an unknown etiology, rapid progression, and a poor prognosis [1]. The diagnosis is made by characteristic radiological and/or histological findings on high-resolution computed tomography and lung biopsy, respectively, in the absence of a specific identifiable cause. 

Rising awareness of the clinical signs of idiopathic pulmonary fibrosis (IPF), along with the broader use of CT scans and other factors, has led to an increased prevalence of IPF over the past 20 years [2]. Significant progress has been made in understanding the pathobiology of IPF, identifying numerous genetic and non-genetic contributors. The disease course and progression rate in IPF patients are often unpredictable and vary widely. The decline in lung function is further influenced by antifibrotic treatments, which have been proven to slow the disease’s progression. Comorbid conditions can heighten symptom severity and affect survival rates. Conditions like pulmonary hypertension, emphysema, lung cancer, and gastroesophageal reflux have been extensively studied [3]. However, there is less information on endocrine disorders. A low body mass index (BMI) and weight loss have been linked to higher mortality rates [4].

Moreover, the progression and outlook of the disease can differ greatly, influenced by acute exacerbations, comorbid conditions, disease severity, and the availability and side effects of antifibrotic medications. Staging systems for disease severity are essential for predicting outcomes and guiding treatment decisions. Various clinical predictive models have been created for patients with IPF, with the Gender, Age, and Physiology (GAP) model being the most commonly used [5].

Therefore, a new nutritional approach is needed that focuses on assessing nutritional status by examining changes in body composition and function. According to the latest diagnostic criteria from the Global Leadership Initiative on Malnutrition (GLIM), malnutrition is identified in individuals who test positive on a screening test and meet at least one phenotypic criterion (such as unintentional weight loss, low muscle mass, or low body mass index) and one etiologic criterion (such as reduced food intake or assimilation or disease burden/inflammatory condition) [6].

Sarcopenia can be diagnosed following the recommendations from the European Working Group on Sarcopenia in Older People 2 (EWGSOP2). These guidelines include three criteria: low muscle strength (which identifies probable sarcopenia and allows for therapeutic intervention), low muscle mass (necessary to confirm the diagnosis), and low physical performance (which determines the severity of sarcopenia) [7].

Few studies have examined the prevalence of sarcopenia or malnutrition in individuals with IPF, and none have applied the latest diagnostic criteria (GLIM and EWGSOP2). Additionally, none of these studies investigated both conditions together.

This analysis aimed to evaluate the prevalence of malnutrition, sarcopenia, and the combined malnutrition-sarcopenia syndrome in individuals with IPF and explore the relationship between IPF and these conditions.

## 2. Methods

### 2.1. Setting of Study

A prospective, observational, multicenter study of routine clinical practice was conducted in the Respiratory and Endocrinology Units at the Virgen de la Victoria University Hospital. Consecutive patients attending an ILD clinic with idiopathic pulmonary fibrosis (IPF) were diagnosed at various stages. Participants were referred from both our hospital and the Regional University Hospital of Málaga.

All participants provided informed consent prior to joining the study. The study adhered to the principles of the Declaration of Helsinki and received approval from the Ethics Committee of Málaga on 5 April 2022. All patients included in the study met the inclusion criteria (diagnosis of idiopathic pulmonary fibrosis and consent to participate) and did not meet any exclusion criteria (refusal to participate or inability to perform BIA measurement due to factors like ethnicity, extensive skin lesions, fluid extravasation, local hematomas, amputation, or severe, active, and/or uncontrolled cardiovascular, musculoskeletal, metabolic, malignant, or neurological conditions that limited function or resulted in a life expectancy of less than three months).

### 2.2. Evaluation of Nutritional Status

Malnutrition was diagnosed based on the GLIM criteria [6]. The diagnosis of malnutrition was confirmed if at least one phenotypic and one etiologic criterion were met, as recommended by the GLIM experts.

Phenotypic criteria:Unintentional weight loss: more than 5% of usual body weight lost within the past six months, or more than 10% lost over more than six months.Low body mass index (BMI): less than 20 kg/m^2^ for individuals under 70 years old, and less than 22 kg/m^2^ for those 70 years or older.Low muscle mass (LMM): ASMM (appendicular skeletal muscle mass measured by BIVA) <20 kg for men and <15 kg for women, or ASMI (appendicular skeletal muscle mass index) <7.0 kg/m^2^ for men and <5.5 kg/m^2^ for women.

Appendicular lean mass was measured using the electrical bioimpedance method with the BIVA 101 Whole Body Bioimpedance Vector Analyzer (AKERN, Florence, Italy).

Etiologic criteria:

Reduced food intake or assimilation was identified in individuals reporting any reduction in food intake within the past three months on the subjective global assessment SGA questionnaire.

4.Disease burden/inflammatory condition was identified in all participants diagnosed with IPF.

### 2.3. Evaluation of Sarcopenia

To diagnose sarcopenia, we used the EWGSOP2 criteria [7,8] of HGS <27 kg for men and <16 kg for women to indicate probable Sc. If it was combined with low muscle mass measured by BIVA, with ASMM (appendicular skeletal muscle mass by BIVA) <20 kg for men and <15 kg for women or ASMI (appendicular skeletal muscle mass index) <7.0 kg/m^2^ for men or <5.5 kg/m^2^ for women, the diagnosis of Sc was made. 

### 2.4. Evaluation of Malnutrition-Sarcopenia Syndrome (MSS)

To establish MSS (malnutrition-sarcopenia syndrome) as a new criterion that allows for early intervention, we propose using it to identify patients at risk before they reach significant weight loss thresholds or advanced stages of sarcopenia and malnutrition. Specifically, MSS would be diagnosed when there is >5% weight loss combined with low muscle mass and function (sarcopenia). Unlike the GLIM criteria, this syndrome includes an assessment of patient functionality through handgrip strength testing, providing a direct measure of muscle strength and, consequently, the functional status of the individual. This approach would enable early detection and intervention, preventing progression to more severe diagnoses of sarcopenia and malnutrition. By identifying at-risk patients sooner, we can initiate nutritional and physical interventions earlier, potentially improving clinical outcomes and quality of life for these individuals.

### 2.5. Respiratory Evaluation 

All subjects enrolled in the study had idiopathic pulmonary fibrosis diagnosed based on the ATS/ERS/JRS/ALAT Clinical Practice Guideline [1]. Information on the results of the spirometry was obtained from the data collected in the pulmonology clinical history. The severity of IPF was staged based on GAP [5]. The GAP composite index was developed in 2012 and is based on the assessment of gender (G), age (A), and physiology (P), the latter including two conventional lung function parameters: forced vital capacity (FVC) and diffusing lung capacity for CO (DLCO). 

Functional tests (Timed Up and Go, TUG) were performed with the patient sitting in a chair and being timed in seconds for how long it took them to get up, walk 3 m, turn around, and walk another 3 m to sit down. The same was done with the 6 min walk, timing how many meters the patient traveled in 6 min to assess their ability [9].

The St. George’s Respiratory Questionnaire (SGRQ) and 12-Item Short Form Health Survey (SF-12) were administered at the visit. The SGRQ is a standardized questionnaire used to assess health-related quality of life in patients with respiratory conditions, particularly chronic obstructive pulmonary disease (COPD) and asthma. It consists of 50 items grouped into three components: symptoms, activity, and impact. The SGRQ has been widely used in clinical trials and research studies to assess treatment outcomes and the effectiveness of interventions in respiratory diseases [10].

The SF-12 is a widely used tool for measuring health-related quality of life across various populations, including those with chronic diseases. It is a shortened version of the SF-36, consisting of 12 questions that evaluate both physical and mental health components. The survey provides two summary scores: the Physical Component Summary (PCS) and the Mental Component Summary (MCS). The SF-12 has been extensively validated and employed in clinical research to assess the overall health status and the impact of different medical conditions and treatments on quality of life.

### 2.6. Statistical Analysis

The data analysis was mainly performed using JAMOVI (version 2.2.2 MacOS). Descriptive statistics were used to characterize the patient cohort. Descriptive statistics were used to analyze the categorical variables (absolute and relative frequencies) and the quantitative variables (means and SDs or medians and interquartile ranges).

The normality of the distribution of quantitative variables was checked using the Shapiro–Wilk test. The clinical data and values between malnutrition and non-malnutrition and sarcopenia and non-sarcopenia were compared using Student’s *t*-test, the Mann–Whitney U test, or the Chi-squared test. A *p*-value of less than 0.05 was considered significant.

## 3. Results

a.General Characterization of the Study Population

A total of 85 patients with IPF were included. All subjects had IPF diagnosed based on the ATS/ERS/JRS/ALAT Clinical Practice Guideline [1]. The mean age was 71 (±7.22 years). A total of 71 patients were men (83.5%), and 14 were women (16.5%). Regarding tobacco use, 12.7% had no history of tobacco use, 4.2% were active smokers, and 83.1% were former smokers.

Regarding mean weight, it was 79.1 kg (±12.5 kg), and the mean height was 169 cm (±8.03 cm). The mean BMI (kg/m^2^) was 27.6 (3.47).

For the variable GAP, the distribution was as follows: GAP I included 28 patients (32.94% of the total), GAP II had 40 patients (47.05% of the total), and GAP III comprised 17 patients (20%of the total); Figure 1.

Regarding the characteristics of the patient’s pulmonary function tests, the mean FVC was 2620 (±777), and the mean DLCO was 49.6 (±17.3). A total of 71 patients were using antifibrotic treatment (22 with Pirfenidone and 49 with Nintedanib).

Regarding the Charlson Comorbidity Index, 8.3% had a score of 2, 22.2% had a score of 3, 40.3% had a score of 4, 18.1% had a score of 5, and 11.1% had a score of 6. 

b.Malnutrition

Malnutrition was diagnosed in 66 patients (77.65%) according to the GLIM criteria. Table 1 presents the characteristics of the variables.

The analysis revealed significant differences between the malnourished and non-malnourished patients. The non-malnourished patients exhibited greater body weight (86.73 ± 10.29 kg, *p* = 0.02), taller stature (172.21 ± 6.33 cm, *p* = 0.044), and higher BMI (29.30 ± 3.52, *p* = 0.017) compared to the malnourished group. No significant differences were found in FVC or DLCO. There were no statistically significant differences in the TUG test or the 6-minute walk test.

Additionally, the non-malnourished patients demonstrated significantly higher values for Appendicular Skeletal Muscle Mass (ASMM) (23.18 ± 2.30 kg) and Appendicular Skeletal Muscle Mass Index (ASMI) (7.81 ± 0.53), with *p*-values less than 0.001 for both measurements. There were significant differences in HGS, with lower grip strength observed in the malnourished patients.

In terms of quality of life, the patients in the malnourished group reported significantly poorer outcomes. They demonstrated worse health-related quality of life scores on the Saint George’s Respiratory Questionnaire (total score: 48.55 ± 25.37) compared to the non- malnourished patients, who exhibited lower scores (28.27 ± 19.10). Across the symptom, activity, and impact domains of the Saint George’s Questionnaire, the risk group consistently presented higher scores, all with *p*-values less than 0.05.

Moreover, physical health, as measured by the SF-12 Physical score, was significantly better in the non-malnourished patients. The non-malnourished patients had a significantly higher physical score (42.44 ± 8.76) compared to those in the malnourished group (35.54 ± 10.72), with a *p*-value of 0.027.

c.Sarcopenia

Sarcopenia was diagnosed in 17 patients (20%); Table 2 shows the characteristics of the subgroups with and without sarcopenia.

The patients who were sarcopenic had a significantly higher age (73.90 ± 7.19), lower weight (72.83 ± 9.87 kg), and lower height (162.26 ± 7.30 cm) compared to the non-sarcopenic patients (170.88 ± 7.20 cm), with a *p*-value of <0.001. No significant differences between DLCO or BMI were found. There were no significant differences in the GAP stage distribution between the sarcopenic and non-sarcopenic patients, with a *p*-value of 0.068.

The sarcopenic patients also had significantly lower ASMM (16.96 ± 1.75) and ASMI_Sarc (6.44 ± 0.46) values compared to the non-sarcopenic patients (21.40 ± 2.98 and 7.31 ± 0.73 respectively), with *p*-values of 0.001 for both measures.

Handgrip strength (HGS) was significantly lower in the sarcopenic patients (20.19 ± 4.39) compared to the non-sarcopenic patients (36.56 ± 7.84), with a *p*-value < 0.001. Timed Up and Go (TUG) performance was significantly worse in the sarcopenic patients (9.27 ± 2.68) compared to the non-sarcopenic patients (7.23 ± 1.71), with a *p*-value of 0.001. No significant differences were observed in the 6 MWT.

The sarcopenic patients had significantly higher scores in the CRSG impact section (57.37 ± 22.58) compared to the non-sarcopenic patients (39.45 ± 24.94), with a *p*-value of 0.016. Significant differences were found for the SF-12 Physical component, but not for the SF-12 Mental component.

d.Malnutrition-Sarcopenia Syndrome (MSS)

Malnutrition-sarcopenia syndrome (MSS) was diagnosed in seven patients (8.23%). Table 3 presents the characteristics of the subgroups with and without malnutrition-sarcopenia. No significant difference was observed in DLCO.

The patients with MSS were significantly older (76.86 ± 5.18 years), with a *p*-value of 0.022, had lower body weight (65.03 ± 7.57 kg, *p* = 0.002), and were shorter in stature (162.71 ± 7.41 cm, *p* = 0.031).

The patients with MSS also exhibited significantly lower body weight (65.03 ± 7.57 kg) compared to those without MSS (80.32 ± 12.13 kg), with a *p*-value of 0.02. Height was also significantly lower in the MSS patients (162.71 ± 7.41 cm) compared to the non-MSS patients (169.51 ± 7.89 cm), with a *p*-value of 0.031.

No significant differences were found in GAP stage distribution between the patients with and without MSS (*p* = 0.302).

The patients with MSS had significantly lower Appendicular Skeletal Muscle Mass (ASMM) (16.74 ± 2.15 kg) and Appendicular Skeletal Muscle Mass Index (ASMI) (6.30 ± 0.48) values compared to those without MSS (20.73 ± 3.21 kg and 7.19 ± 0.75, respectively), with *p*-values of <0.002 and <0.003 for both measures. Handgrip strength (HGS) was also significantly lower in the MSS patients (23.71 ± 2.98 kg) compared to the non-MSS patients (33.72 ± 9.94 kg), with a *p*-value of 0.005. However, no significant difference was observed in Timed Up and Go (TUG) performance between the MSS patients (8.17 ± 1.72 seconds) and the non-MSS patients (7.65 ± 2.17 seconds), with a *p*-value of 0.386.

The patients with MSS had significantly higher scores in the CRSG impact section (60.22 ± 21.14) compared to those without MSS (41.66 ± 25.30), with a *p*-value of 0.071. Furthermore, the MSS patients showed significantly lower scores in the SF-12 Physical section (30.54 ± 9.38) compared to the non-MSS patients (38.02 ± 10.57), with a *p*-value of 0.044.

## 4. Discussion

Malnutrition and changes in body composition are well established in chronic lung diseases; however, research on nutritional status in interstitial lung disease (ILD) remains limited [11,12]. As far as we know, this is the first study to evaluate the prevalence of malnutrition, sarcopenia, and malnutrition-sarcopenia syndrome in individuals with IPF using the latest diagnostic criteria (GLIM and EWGSOP2).

Both malnutrition and sarcopenia are common in patients with IPF [13,14]. Many factors contribute to reduced muscle mass in IPF patients, including aging, smoking, reduced caloric intake, increased respiratory muscle demand, reduced physical activity, systemic inflammation, and the side effects of steroid and anti-fibrotic treatments [15].

It should be noted that IPF itself fulfills one of the etiologic GLIM criteria for malnutrition (disease burden/inflammatory condition). Therefore, all subjects with IPF should be screened for malnutrition and sarcopenia.

Three out of four participants in our study population had malnutrition, and one out of five had sarcopenia. It should be emphasized that both conditions were diagnosed in 8.23% of the subjects.

We also explored the relationship between nutritional status and IPF severity. Although no statistically significant differences were found in FVC or DLCO between the malnourished and non-malnourished patients, we observed that the majority of malnourished individuals were classified as GAP stage II or III. This suggests a potential association between more advanced disease and nutritional deterioration, which deserves further investigation.

The prevalence of malnutrition and sarcopenia in individuals with idiopathic pulmonary fibrosis (IPF) has been explored in only a limited number of studies [15], with the findings showing variability depending on the assessment tools used. A recent study by Qinxue Shen et al. [16] demonstrated significant differences in malnutrition prevalence based on the scoring systems applied: 37.5% using the Prognostic Nutritional Index, 47.4% with the Controlling Nutritional Status score, and 6.4% with another prognostic nutritional tool. Other studies have reported malnutrition rates ranging from 18.5% to 55% [14,17], highlighting the need to standardize nutritional assessment methods. In our work, we propose the use of the GLIM criteria as a standardized approach to address these inconsistencies.

Importantly, our results demonstrate that malnutrition and sarcopenia were associated with poorer patient-centered outcomes, particularly in health-related quality of life. Both groups presented significantly worse scores in the Saint George’s Respiratory Questionnaire and in the SF-12 Physical component, highlighting the clinical relevance of nutritional status beyond conventional lung function metrics.

The high malnutrition prevalence of 77.65% found in our study can be explained by several factors. First, other studies assessing nutrition in patients with interstitial lung diseases have often included patients with various interstitial pathologies beyond IPF, which may account for lower reported rates of malnutrition. In contrast, our focus exclusively on IPF patients likely contributed to the higher prevalence observed. Additionally, the GLIM criteria may be more sensitive in detecting malnutrition [17]. Another factor that could explain our findings is the severity of the disease in our sample, as most patients were classified as GAP stage II or III.

In an exploratory analysis, no significant association was found between antifibrotic use (including Nintedanib and Pirfenidone) and malnutrition or sarcopenia in our cohort. 

Sarcopenia is a condition characterized by a reduction in both muscle mass and physical performance. While it is well recognized as a complication of chronic diseases like chronic obstructive pulmonary disease (COPD), its presence and impact in patients with idiopathic pulmonary fibrosis (IPF) are less understood. Studies have reported sarcopenia prevalence rates in patients with IPF ranging between 22.9% and 39.3% [13,18]. Our finding of a 20% prevalence is consistent with these figures from the literature.

Malnutrition plays a significant role in the development of sarcopenia by limiting the caloric and protein intake necessary to maintain muscle mass. Conversely, sarcopenia can exacerbate malnutrition by reducing mobility, making it more difficult for patients to shop, cook, and consume adequate nutrition [19]. 

The MSS proposed in this study offers a more comprehensive approach for the early detection of malnutrition and sarcopenia compared to the conventional GLIM criteria. Unlike the GLIM criteria, MSS is not solely based on body mass indicators and weight loss but also incorporates a measure of patient functionality through handgrip dynamometry. 

This functional component is essential, as handgrip strength directly assesses muscle capacity, which is a key indicator of the patient’s functional status that the GLIM criteria do not explicitly include. Previous studies [20,21,22] have shown that loss of functionality often precedes significant weight loss in patients with chronic diseases, suggesting that including a functional measure like dynamometry could enable earlier interventions.

The relationship between malnutrition and sarcopenia is closely interconnected, with both conditions influencing each other [23]. In this research, malnutrition-sarcopenia was diagnosed in 8.23% of the patients.

Therefore, the use of MSS allows for the identification of patients in the early stages of deterioration before they reach higher weight loss thresholds or advanced manifestations of sarcopenia and malnutrition. This early identification is critical, as it opens the door to proactive nutritional and physical interventions that could slow down or even reverse muscle deterioration and improve patients’ quality of life. Additionally, MSS could be especially beneficial in populations with chronic conditions such as idiopathic pulmonary fibrosis (IPF), where malnutrition and sarcopenia are common but often underdiagnosed. Our findings suggest that implementing MSS as a diagnostic criterion could enhance detection and allow for more effective clinical management, thereby optimizing health outcomes and prognosis in this population.

This study included IPF patients with common characteristics of this disease, as they were mostly men. An important finding is that despite the mean BMI indicating overweight (BMI 27.6 kg/m^2^), it is important to highlight that BMI alone is not sufficient to diagnose nutritional disorders in these patients. Our results indicate that BMI is not a reliable predictor of sarcopenia. Also, malnourished patients or those with MSS frequently exhibit BMI values that could misleadingly categorize them as overweight. Although weight loss has indeed been shown to be a predictor of survival [24], our findings suggest that BMI alone is inadequate for accurately assessing patients with IPF, reinforcing the need for more comprehensive evaluation methods to assess nutritional status and muscle mass in this population.

Other studies have examined the relationship between malnutrition and mortality in IPF patients by measuring BMI or weight loss, but BMI provided contradictory results [25]. Although a BMI <21 kg/m^2^ has been used to define malnutrition in patients with chronic diseases, such as COPD [26], versus BMI <25 for patients with IPF [27], it was not sensitive enough to identify patients with a low muscle mass (FFM) in these studies.

Different studies have shown how body composition impacts the quality of life and prognosis in idiopathic pulmonary fibrosis (IPF) [14,17,27,28]. We found lower Appendicular Skeletal Muscle Mass (ASMM) and Appendicular Skeletal Muscle Mass Index (ASMI) values in the malnourished and sarcopenic patients, highlighting the importance of muscle preservation in this population. Muscle loss can directly affect functional capacity and quality of life, which aligns with our findings that both malnourished and sarcopenic patients had poorer scores on the St. George’s Respiratory Questionnaire, establishing a clear relationship between worse nutritional status and lower quality of life.

The GAP index does not predict the risk of nutritional impairment; therefore, patients in the early stages, including those with GAP stage I, should also undergo nutritional evaluation.

A very interesting finding from our study is that both the malnourished and the sarcopenic patients with idiopathic pulmonary fibrosis (IPF) had lower handgrip strength (HGS). This is a simple measure that can be easily assessed during routine consultations, and we propose its use as a straightforward screening tool to detect nutritional alterations in our patients. Regarding the 6-minute walk test, the results were not significant; however, not all patients underwent this test. This was because the patients with more advanced IPF, who were suspected to be more malnourished, did not have this test recorded.

Although malnutrition and sarcopenia are known to have negative consequences and are prevalent among individuals with idiopathic pulmonary fibrosis (IPF), both conditions are often underdiagnosed in this population. This lack of diagnosis hinders the initiation of appropriate therapeutic strategies. Our findings support the need for routine screening of malnutrition and sarcopenia in all patients with IPF, regardless of disease severity. Implementing such an approach would require the involvement of specialists in endocrinology and nutrition in our interstitial pathology committees, which could improve patient outcomes and positively impact prognosis.

Finally, although malnutrition and sarcopenia have been associated with worse prognosis in other chronic diseases, our cross-sectional design did not allow us to evaluate their impact on clinical progression or mortality in patients with IPF. Future longitudinal studies are needed to investigate the prognostic implications of nutritional status in this population.

To conclude, this study has several limitations. First, the sample size was relatively modest, which may affect the generalizability of the findings. Secondly, the lack of multivariate analyses is a limitation of our study, while our study highlights the importance of the fact that this is the first analysis of the prevalence of both malnutrition and sarcopenia in older patients with IPF, in which the most recent diagnostic criteria have been used. It should be emphasized that patients with IPF are at particular risk for both conditions.

## 5. Conclusions

This study highlights the high prevalence of malnutrition and sarcopenia in patients with idiopathic pulmonary fibrosis (IPF) and emphasizes the importance of early nutritional assessment in this population. The proposed malnutrition-sarcopenia syndrome (MSS) offers a more comprehensive diagnostic criterion than conventional tools, incorporating a functional assessment through handgrip dynamometry. This early identification facilitates timely nutritional and physical interventions, with the potential to improve clinical outcomes, quality of life, and prognosis. Our findings support the adoption of MSS as a valuable diagnostic tool in the clinical management of chronic conditions such as IPF, where traditional metrics like BMI or GLIM criteria may fail to detect early nutritional impairments.

## Figures and Tables

**Figure 1 arm-93-00011-f001:**
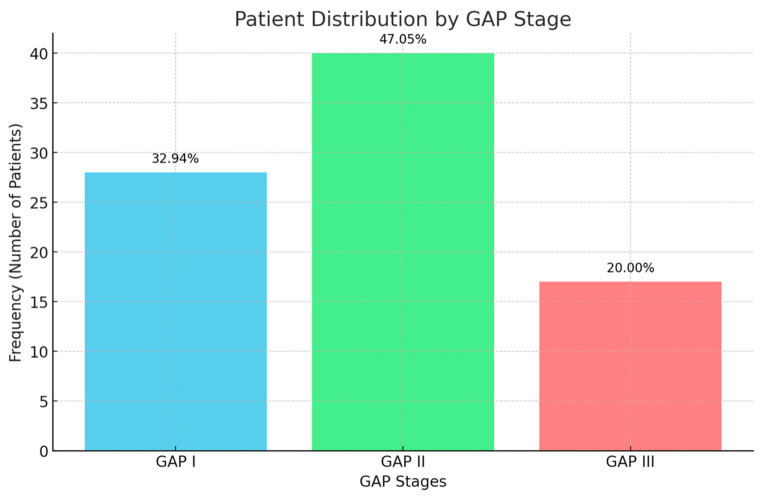
The figure shows the distribution of patients according to GAP stage. Abbreviations: GAP, gender (G), age (A), and physiology (P) index.

**Table 1 arm-93-00011-t001:** Characteristics of subgroups with and without malnutrition. Data are expressed as means plus standard deviations or percentages. Abbreviations: BMI, body mass index; ASMM, appendicular skeletal mass muscle; ASMI, skeletal mass index; HGS mean, mean hand grip strength; TUG, Timed Up and Go test; 6 MWT, 6 min walk test; FVC, forced vital capacity; DLCO, pulmonary carbon monoxide diffusing capacity; GAP, gender (G), age (A), and physiology (P) index.

Variable	Malnutrition (n = 66)	Non-Malnutrition (n = 19)	*p*-Value
**Age (years)**	71.08 ± 7.13	70.90 ± 7.72	0.924
**Weight (kg)**	76.86 ± 12.29	86.73 ± 10.29	**0.002**
**Height (cm)**	168.02 ± 8.27	172.21 ± 6.33	**0.044**
**BMI (kg/m^2^)**	27.16 ± 3.33	29.30 ± 3.52	**0.017**
**ASMM (kg)**	19.61 ± 3.13	23.18 ± 2.30	**<0.001**
**ASMI_Sarc**	6.91 ± 0.71	7.81 ± 0.53	**<0.001**
**HGS (kg)**	31.66 ± 10.15	37.21 ± 7.99	**0.031**
**TUG (s)**	7.87 ± 2.28	7.08 ± 1.40	0.105
**6 MWT (m)**	410.68 ± 72.22	411.25 ± 36.06	0.602
**GAP I**	26.8%	5.6%	0.198
**GAP II**	33.8%	14.1%	
**GAP III**	18.3%	1.4%	
**DLCO (%)**	49.06 ± 18.01	51.64 ± 15.20	0.628
**KCO (%)**	77.04 ± 21.55	85.92 ± 19.22	0.124
**FVC (%)**	2539.76 ± 775.36	2884.44 ± 744.48	0.100
**CRSG tot**	48.55 ± 25.37	28.27 ± 19.10	**0.006**
**CRSG symptoms**	37.66 ± 20.00	25.75 ± 15.21	**0.038**
**CRSG act**	74.81 ± 30.85	54.61 ± 43.04	**0.049**
**CRSG imp**	36.85 ± 28.94	14.03 ± 12.55	**0.004**
**SF-12 Physical**	35.54 ± 10.72	42.44 ± 8.76	**0.027**
**SF-12 Mental**	45.72 ± 13.96	42.06 ± 13.25	0.372

**Table 2 arm-93-00011-t002:** Characteristics of subgroups with and without sarcopenia. Data are expressed as means plus standard deviations or percentages. Abbreviations: BMI, body mass index; ASMM, appendicular skeletal mass muscle; ASMI, skeletal mass index; HGS mean, mean hand grip strength; TUG, Timed Up and Go test; 6 MWT, 6 min walk test; FVC, forced vital capacity; DLCO, pulmonary carbon monoxide diffusing capacity; GAP, gender (G), age (A), and physiology (P) index.

Variable	Sarcopenic (n = 17)	Non-Sarcopenic (n = 67)	*p*-Value
**Age (years)**	73.90 ± 7.19	70.21 ± 7.07	**0.049**
**Weight (kg)**	72.83 ± 9.87	80.86 ± 12.69	**0.013**
**Height (cm)**	162.26 ± 7.30	170.88 ± 7.20	**<0.001**
**BMI (kg/m^2^)**	27.71 ± 3.71	27.61 ± 3.43	0.919
**ASMM (kg)**	16.96 ± 1.75	21.40 ± 2.98	**<0.001**
**ASMI_Sarc**	6.44 ± 0.46	7.31 ± 0.73	**<0.001**
**HGS (kg)**	20.19 ± 4.39	36.56 ± 7.84	**<0.001**
**TUG (s)**	9.27 ± 2.68	7.23 ± 1.71	**<0.001**
**6 MWT (m)**	384.60 ± 78.77	417.05 ± 61.16	0.160
**GAP I**	12.7%	19.7%	0.068
**GAP II**	7.0%	40.8%	
**GAP III**	2.8%	16.9%	
**DLCO (%)**	54.07 ± 24.46	48.23 ± 14.41	0.260
**KCO (%)**	78.93 ± 26.36	79.02 ± 19.53	0.989
**FVC (%)**	2185.00 ± 796.61	2753.15 ± 727.41	**0.006**
**CRSG tot**	57.37 ± 22.58	39.45 ± 24.94	**0.016**
**CRSG symptoms**	41.35 ± 16.69	32.79 ± 20.07	0.140
**CRSG act**	83.08 ± 19.88	65.91 ± 37.59	0.096
**CRSG imp**	47.36 ± 30.07	26.44 ± 25.20	**0.009**
**SF-12 Physical**	31.84 ± 9.93	38.85 ± 10.39	**0.025**
**SF-12 Mental**	49.58 ± 14.69	43.37 ± 13.29	0.129

**Table 3 arm-93-00011-t003:** Characteristics of subgroups with and without malnutrition-sarcopenia syndrome. Data are expressed as means plus standard deviations or percentages. Abbreviations: BMI, body mass index; ASMM, appendicular skeletal mass muscle; ASMI, skeletal mass index; HGS mean, mean hand grip strength; TUG, Timed Up and Go test; 6 MWT, 6 min walk test; FVC, forced vital capacity; DLCO, pulmonary carbon monoxide diffusing capacity; GAP, gender (G), age (A), and physiology (P) index.

Variable	MSS (n = 7)	Non-MSS (n = 78)	*p*-Value
**Age (years)**	76.86 ± 5.18	70.51 ± 7.17	**0.022**
**Weight (kg)**	65.03 ± 7.57	80.32 ± 12.13	**0.002**
**Height (cm)**	162.71 ± 7.41	169.51 ± 7.89	**0.031**
**BMI (kg/m^2^)**	24.59 ± 2.63	27.91 ± 3.42	**0.015**
**ASMM (kg)**	16.74 ± 2.15	20.73 ± 3.21	**0.002**
**ASMI_Sarc**	6.30 ± 0.48	7.19 ± 0.75	**0.003**
**HGS (kg)**	23.71 ± 2.98	33.72 ± 9.94	**0.005**
**TUG (s)**	8.17 ± 1.72	7.65 ± 2.17	0.386
**6 MWT (m)**	423.75 ± 49.56	409.73 ± 66.78	0.891
**GAP I**	4.2 %	28.2 %	0.302
**GAP II**	2.8 %	45.1 %	
**GAP III**	0.0 %	19.7 %	
**DLCO (%)**	68.00 ± 22.73	48.04 ± 16.05	**0.049**
**KCO (%)**	93.20 ± 25.29	77.68 ± 20.59	0.072
**FVC (%)**	2663.33 ± 894.44	2616.70 ± 773.83	0.842
**CRSG tot**	60.22 ± 21.14	41.66 ± 25.30	**0.071**
**CRSG symptoms**	43.45 ± 19.51	33.75 ± 19.44	0.255
**CRSG act**	79.97 ± 20.49	68.75 ± 36.19	0.764
**CRSG imp**	53.38 ± 25.24	28.67 ± 26.93	**0.022**
**SF-12 Physical**	30.54 ± 9.38	38.02 ± 10.57	**0.044**
**SF-12 Mental**	52.78 ± 14.28	43.86 ± 13.52	0.123

## Data Availability

The original contributions presented in the study are included in the article, further inquiries can be directed to the corresponding authors.

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
