# Peer review of "Malnutrition, Sarcopenia, and Malnutrition Sarcopenia Syndrome in Idiopathic Pulmonary Fibrosis"

_arm, 2025, doi:10.3390/arm93030011_

Round 1

Reviewer 1 Report

Comments and Suggestions for Authors

Thank you for the opportunity to review this manuscript. The subject matter is interesting and potentially relevant.

The major issue I have at the moment is that sarcopenia and malnutrition are being treated as seperate entities from the disease process, even though I think they may be related to the disease process, with more severe disease being associated with increased sarcopenia and malnutrition. The most important question, then, is whether adequately treating malnutrition and sarcopenia can influence prognosis. To further elaborate on this the following can be done:

  • indicate more clearly whether sarcopenia/malnutrition was associated with more severe IPF in terms of lung function impairment and GAP-stage
  • indicate whether sarcopenia/malnutrition was associated with outcomes
  • indicate whether sarcopenia/malnutrition was associated with nintedanib use (as this can give diarrhea as a side effect)

Minor comments:

  • line 330-331 is in Spanish and should be translated into English

Author Response

Response to Reviewer 1

We sincerely thank you for your valuable comments and the interest shown in our work. Below, we address each of your points and have made the corresponding changes in the manuscript:

  1. Association between malnutrition/sarcopenia and IPF severity: We have expanded the discussion to more thoroughly explore the potential association between these conditions and disease severity. Although no statistically significant differences were found in FVC or DLCO between malnourished and non-malnourished patients, we observed that most malnourished individuals were classified as GAP stage II or III. This suggests a possible correlation between more severe disease and nutritional deterioration. This observation has been incorporated and discussed in the revised manuscript.
  2. Association with outcomes: We have included in the manuscript that both malnutrition and sarcopenia were associated with poorer health-related quality of life, as measured by the St. George's Respiratory Questionnaire and the SF-12 physical component. These associations are detailed in the results section and emphasized in the discussion as clinically relevant outcomes. Although the study was not designed to evaluate mortality, we reference previous studies that have established this association, thereby reinforcing the clinical relevance of our findings.
  3. Association with nintedanib use: As you correctly point out, nintedanib can be associated with weight loss due to gastrointestinal side effects. We reviewed our data and found no statistically significant association between nintedanib use and the presence of malnutrition or sarcopenia. This exploratory analysis has been added to the results and briefly discussed in terms of potential clinical implications. 

Minor comments:

  • line 330-331 is in Spanish and should be translated into English: Done.

Once again, we thank you for your insightful suggestions, which have helped improve the quality and clarity of our manuscript.

Reviewer 2 Report

Comments and Suggestions for Authors

In this cohort study of 85 patients with IPF, the authors assessed the prevalence of malnutrition and sarcopenia. It was found that malnutrition occurred in 80% of patients and sarcopenia in 20%, and patients with of malnutrition and sarcopenia had worse health-related quality of life.

The study had a relatively simple and straightforward design, and overall, it was presented clearly and concisely.

I have a number of concerns and comments.

Unfortunately, the authors identified only one negative consequence of of malnutrition and sarcopenia - worse quality of life. It would have been interesting to demonstrate the its effect on progression of disease and prognosis as well.

In addition, given that most patients were taking antifibrotics, it would have been interesting to assess the effects of this therapy on of malnutrition and sarcopenia, and to devote space to this issue in the discussion.

Minor comments:

No data on smoking status of patients

Comments on the Quality of English Language

In several places the text is not in English:

-330-331: y uno de cada cinco tiene sarcopenia

-393: Aunque la perdida de peso si se ha demostrado que es un

-Table 1-3: sintom

FCV

Author Response

We sincerely appreciate your thoughtful comments and your recognition of the clarity and relevance of our study. Below we provide responses to the issues you raised:

  1. Impact of malnutrition and sarcopenia on disease progression and prognosis:
    We agree that evaluating the relationship between nutritional status and disease progression or prognosis in IPF would be highly valuable. However, as this was a cross-sectional observational study with no longitudinal follow-up, we were not able to assess outcomes such as progression, hospitalization, or mortality. Nevertheless, we have now expanded the discussion to emphasize this limitation and to suggest the need for future prospective studies designed to evaluate the prognostic implications of malnutrition and sarcopenia in IPF patients.
  2. Effect of antifibrotic therapy on malnutrition and sarcopenia:
    As you correctly pointed out, antifibrotic therapy—particularly nintedanib—may influence nutritional status due to its gastrointestinal side effects. In response to your suggestion, we performed an exploratory analysis and found no statistically significant association between the use of antifibrotic therapy and the presence of malnutrition or sarcopenia in our cohort. These results, as well as their clinical implications, have now been included and discussed in the revised manuscript.

Minor comments:

No data on smoking status of patients : Additionally, regarding your implicit interest in patient characteristics such as smoking history, we would like to note that this information is provided in the “General Characterization of the Population Study” section of the results. Specifically, we report the distribution of tobacco use, indicating that 12.7% of patients had no history of smoking, 4.2% were current smokers, and 83.1% were former smokers.

Comments on the Quality of English Language: done

Once again, we thank you for your constructive feedback, which has allowed us to further improve the scope and quality of our manuscript.

Round 2

Reviewer 1 Report

Comments and Suggestions for Authors

Thank you for addressing my prior remarks. I do not have any further comments now.

Reviewer 2 Report

Comments and Suggestions for Authors

The authors have responded to all of the reviewer's comments, and the paper may be accepted for publication.